# Assessing the Level of Knowledge and Experience Regarding Cervical Cancer Prevention and Screening among Roma Women in Romania

**DOI:** 10.3390/medicina59101885

**Published:** 2023-10-23

**Authors:** Septimiu Voidăzan, Alexandra Mihaela Budianu, Rozsnyai Florin Francisc, Zsolt Kovacs, Cosmina Cristina Uzun, Bianca Elena Apostol, Reka Bodea

**Affiliations:** 1Department of Epidemiology, George Emil Palade University of Medicine, Pharmacy, Science and Technology of Târgu Mureș, 540139 Târgu Mureș, Romania; septimiu.voidazan@umfst.ro (S.V.); bodeareka@gmail.com (R.B.); 2Department of Obstetrics Gynecology, George Emil Palade University of Medicine, Pharmacy, Science and Technology of Târgu Mureș, 540139 Târgu Mureș, Romania; francisc76@gmail.com; 3Department of Biochemistry and Environmental Chemistry, George Emil Palade University of Medicine, Pharmacy, Science and Technology of Târgu Mureș, 540139 Târgu Mureș, Romania; zsolt.kovacs@umfst.ro (Z.K.); comina20uzun@gmail.com (C.C.U.); 4General Medicine, George Emil Palade University of Medicine, Pharmacy, Science and Technology of Târgu Mureș, 540139 Târgu Mureș, Romania; apostolbia2016@gmail.com

**Keywords:** Roma women, screening, prevention, cervical cancer, HPV

## Abstract

*Background and Objectives*: Romania ranks among the countries with a particularly high rate of mortality that can be prevented through prevention programs, screening, early detection, and prompt care. Cervical cancer (CC) is a major cause of these preventable deaths, affecting individuals from marginalized and rural regions, as well as the Roma population. The purpose of this article was to identify accurate and consistent information about the Roma population on the risk of CC, as well as the importance of understanding the causes of the disease and awareness of the available prevention methods. *Materials and Methods*: A cross-sectional study was conducted using a self-administered questionnaire applied only to Roma women in Romania. *Results*: We enrolled 759 patients in this study. These were divided into two groups: Group 1 comprised 289 (38.1%) women who had been tested for HPV infection, while Group 2 included 470 (61.9%) women who had never been tested for HPV infection. Characterization of women in Group 1: mostly aged between 25 and 54 years, with high school education, married, who started sexual activity under the age of 18 years, with only one sexual partner, and had over five pregnancies. Regarding contraceptive methods, 35.7% of women do not know or use any contraceptive method, and 32.2% use hormonal contraceptives. Two thirds of the women tested had heard of HPV, and 19.7% were vaccinated against HPV with at least 2–3 doses. A percentage of 8.7 had a diagnosis of CC, compared to those who were not tested (*p*-0.0001), whereas 63% of the tested women did not know much about CC, as opposed to 85.7% of the group of untested women. *Conclusions*: Cervical cancer (CC) continues to be a public health concern in Romania, particularly among vulnerable groups. Promoting campaigns to raise awareness for HPV vaccination and CC screening are necessary to reduce the associated mortality and morbidity.

## 1. Introduction

Cervical cancer (CC) is the fourth most common type of cancer in women worldwide and the second most common in Romania [1]. The incidence of CC and mortality rates in Romania are three times higher than in other European countries at 22.6 and 9.6 per 100,000 women, respectively [1,2]. According to current estimates, 3380 women in Romania are diagnosed with CC each year, with 1,805 dying as a result of this condition [3].

Despite recent improvements, the health of Romanian populations remains low, with a life expectancy of 75.3 years in 2016, compared to the European Union (EU) average of 81 years. Moreover, the life expectancy of Romanian citizens belonging to the Roma minority is on average 6 years lower than that of the rest of the non-Roma population in the country [4,5].

The healthcare system faces numerous challenges. Low funding and inefficient use of public resources limit the effectiveness of the healthcare system. At the same time, access to adequate healthcare services is still difficult for groups of the female population in Romania, especially those belonging to poor or vulnerable households, rural settlements, and small towns, as well as the Roma population.

In Europe, the Roma are the largest transnational minority, currently reaching 11 million, with the vast majority living in Central and Eastern Europe [6]. In Romania, according to the 2021 census, the Roma accounted for 569,500 people (i.e., 3.4% of the total population), being the second largest minority ethnic group in Romania. Of these, over 60% live in rural areas [7]. There are perspectives suggesting that the estimated figure may not accurately reflect reality since many individuals may be unwilling to disclose their ethnicity. According to sociological studies, the actual Roma population in Romania is believed to range between 1.5 and 2 million people [8]. In addition, Romania has one of the highest rates of potentially avoidable deaths under the conditions of prompt medical assistance and effective application of prevention programs [4].

A clinical area affected by the inefficiency of the system is represented by chronic diseases that are equally tributary to late diagnosis (failure of early detection, absence of active screening) along with the poor control of known cases, such as CC, with the organization of regional screening programs being a necessity derived from the extremely unfavorable epidemiological profile of this malignant condition among women in Romania [4].

Among the possible causes that lead a cancer with curable potential to have such a high impact on mortality in women in Romania, we can list the following: low awareness of the risk of disease, lack of HPV vaccination, suboptimal screening for CC, late presentation to the doctor, fear of a bad diagnosis, primary diagnosis of CC established in advanced stages of disease, reduced opportunities for access to diagnosis and treatment, low number of medical oncologists, and lack of multidisciplinary teams for CC management.

In 2012, the Romanian Ministry of Health implemented a national program that provided for the testing of women aged 25–64 years, at five-year intervals, through the conventional cytological smear. Coverage of the regional pilot program was 21% for all ethnic groups by the end of 2008 [9].

Industrialized nations have reviewed screening programs, recommending HPV testing for women over 30 years of age as an alternative to the cyto-vaginal smear, as well as a progressive transition to HPV testing due to its high effectiveness compared to cytology [10,11,12]. Currently, Romanian national projects [13], in the long term, aim to reduce mortality through CC and change population behaviors by promoting health-friendly ones and reducing those that increase the risk of disease; prolonging the duration of active life in which people can carry out their professional activity and perform family duties independently; decreasing the incidence of CC; decreasing the severity of CC cases; reducing the number and severity of their complications; reducing the need for hospital care, their costs, and disabilities caused by complications; increasing the demand for preventive services; and developing an expectancy to receive such services and not only curative ones.

All these projects have been initiated in direct response to the identified needs of vulnerable groups, including Roma communities, and aim to address the following key issues: improved access to quality healthcare services, especially for residents in rural areas and marginalized groups who often have restricted healthcare access, which has a negative impact on the health of the population; reduction in the number of CC cases—through the detection of precancerous lesions, the screening procedures have the potential to significantly decrease or even eliminate the risk of developing CC; fewer cases of advanced-stage cancer—by detecting cancer in its early phases, before it has a chance to metastasize, these programs can substantially reduce the number of cases of advanced cancer; decreased cancer mortality—early-stage cancer benefits from more effective treatments with higher chances of cure, and detecting cancer in its early stages significantly lowers the risk of cancer-related deaths.

Within various ethnic groups, including the Roma population, there is a lower rate of participation in screening activities compared to the broader Romanian population [14]. This tendency can be attributed to the fact that many Roma women are unaware of the existence of the screening program, have reservations about its cost-free nature, and hold skepticism regarding the potential health benefits of participating in such screenings [15].

Starting from the hypothesis that far less research has been conducted on Roma women in Romania compared to non-Roma women [3,16,17,18,19,20], the purpose of this study was to determine the accuracy and consistency of information regarding the risk of CCC within the Roma population. Furthermore, it aimed to emphasize the need of comprehending the underlying causes of the disease and raising awareness regarding the available prevention methods.

## 2. Materials and Methods

A cross-sectional study was conducted using a self-administered online questionnaire exclusively for the Roma women in Romania. The questionnaire included 31 items with questions about (1) demographic information, (2) information about sex life, (3) knowledge about HPV infection, (4) knowledge about CC, (5) attitude to the prevention of genital tract infections, (6) knowledge of the vaccine that prevents HPV infection, (7) evidence for carrying out the cervico-vaginal cytological examination. Most of the questions were of open type, closed with ordered answers, closed with unordered answers, but also binary questions, with a maximum required time of 15 min to complete. For each question, there was a request for either a single answer or the possibility of multiple answers.

The questionnaire header included information about the scientific purpose of this study, and the anonymity and confidentiality of responses, and by completing the questionnaire questions, respondents implicitly expressed their informed consent. We only considered questionnaires that were completed in full. The sampling was carried out using a conventional non-randomized method. The survey was conducted between January and August 2023, using a Google Form questionnaire, with online application nationwide. The first question in the questionnaire (Are you of Roma ethnicity?) was the basis of the inclusion and exclusion criteria from this study. Inclusion criteria for study participants were women aged between 16 and 64 years, self-identified Roma women, previous history of screening test for HPV infection (yes or no), place of residence urban or rural. Exclusion criteria were other ethnic groups, people under 16 or over 64 years (Figure 1). This study was approved by the Ethics Committee of the University of Medicine and Pharmacy (no. 205 of 21 June 2019).

### Statistical Analysis

Statistical analysis was performed using the Statistical Package for Social Sciences (SPSS, version 23, Chicago, IL, USA). χ^2^ test was used for categorical variables (expressed by numbers (%). Bivariate analysis (chi square tests) was used to assess the associations between each of the independent characteristics and the different variables of interest. All tests were interpreted in relation to statistical significance *p* = 0.05, and statistical significance was considered for *p*-values under the significance threshold.

## 3. Results

### 3.1. Demographic Characteristics of the Study Group

For statistical analysis of the data, we divided the 759 women surveyed in this study into two groups: Group 1 comprised 289 (38.1%) women who had been tested for HPV infection, and Group 2 comprised 470 (61.9%) women who had never been tested for HPV infection. Most tests were conducted in the 35–54 age range (46.1%). Most of the respondents had secondary education (53.2%), and of those with tests carried out, 51.2% had high school education. By marital status, 69.9% of married women had performed an HPV test, but many single women had not. Most of the respondents do not consume alcohol or smoke; however, among those who have been tested for HPV, 18.7% smoke an average of 11–20 cigarettes a day and 18.3% use alcohol 1–2 times a week. Mixed nutrition predominates in women who have had tests performed (Table 1).

### 3.2. Features of the Group According to Sexual Life

Of the respondents, 738 had started their sex life; of these, 572 had started it under the age of 18. About 2/3 of the people in Group 1 began their sex life under 18 years of age. The majority have only one sexual partner and, regarding the frequency of vaginal sex, 90.3% of the women tested disclosed under three intercourses per week. According to pregnancies/miscarriages, 36.3%—21.6% of the tested people—had over 5 pregnancies and over 3 abortions, compared to 30.6%—18.1% of the untested women—who had over 2–3 pregnancies as well as an abortion. Regarding the contraceptive methods used, 35.7% of Group 1 women do not know of or use any contraceptive methods, and 32.2% use hormonal contraceptives. Of the women who have never been tested for HPV infection, 63.8% do not know of/use any contraceptive method (Table 2).

### 3.3. Information about CC, Prevention

Table 3 provides information about the history of CC, as well as about their families, gynecological examinations, treatments, and prevention. Of the women who had ever been tested for HPV, 8.7% had a diagnosis of CC. Within the family of these women, 17.3% have a family member with a diagnosis of CC, while 28.4% have a family member with another type of cancer. All subjects who have been tested for HPV have undergone lifelong gynecological examinations, with 75.4% in the past year. Two thirds of the women tested had heard of HPV, and 19.7% were vaccinated against HPV with at least 2–3 doses of the available vaccine. Of the women tested, most have had a cervico-vaginal cytological examination; in 3.8% of them, cancerous cells were found, and in 2.4%, HSIL- or LSIL-type cells were found. The question “Have you ever had a genital tract infection?” was answered positively by 62.6% of the women tested, and, apart from HPV, they reported candidiasis and urinary tract infections. Of the women tested, 29.4% had had surgery for gynecological diseases compared to 22.8% of the group of untested women (Table 3).

## 4. Discussion

Lifestyle, education (including health education), and access to healthcare services, particularly preventive care, all have a substantial impact on a population’s general health. From this standpoint, prioritizing disease prevention and health maintenance is deemed more effective than disease treatment. However, it is noteworthy that individuals residing in rural areas and vulnerable communities often face restricted access to healthcare services, which invariably has adverse repercussions on the overall health status of the population.

The primary objective of our study was to gauge the level of accurate and consistent information within the Roma population regarding the risk of CC. Specifically, we aimed to understand the extent to which Roma women prioritize comprehending the underlying causes of the disease and familiarizing themselves with the available prevention methods.

The 759 participants included in this study were divided into two groups based on whether or not they had previously been tested for HPV. Group 1 included women who had been tested for HPV, women mostly between the ages of 25 and 54, with high school education, married, who had started sexual activity under 18 years of age, with only one sexual partner, and had over five pregnancies. Among the respondents with performed tests, 18.7% smoked an average of 11–20 cigarettes per day and 18.3% consumed alcohol 1–2 times each week. Women in Group 2, who had never been tested for HPV, were typically younger, under 34, with secondary education, unmarried, under 18 years old, with only one sexual partner, and had between 2 and 5 pregnancies.

Health status indicators highlight a significant disparity in the health of the Roma population when compared to the general population. Notably, Roma women experience more frequent health issues at earlier stages of life. Moreover, the concern arises as a considerable number of Roma respondents lack medical insurance or remain uncertain about their insurance coverage status [15,21,22].

CC stands out as the most preventable type of cancer with an 80% preventability rate achievable through regular testing in organized screening programs. Vaccination against persistent HPV infection and the screening of adult women to detect precancerous lesions of CC are the optimal protection strategies against CC.

Given Romania’s alarming second-place ranking in Europe for a significantly high number of CC-related deaths, it becomes evident that there is a significant lack of public education regarding HPV infection, its role in CC development, and the efficacy of HPV vaccination as a preventive measure. The inadequate participation in screening programs and parental resistance to vaccination serve as indicators of limited awareness regarding HPV infection and its long-term consequences. This situation elevates the significance of addressing HPV infection as a matter of national concern, exerting a profound impact on women’s morbidity and mortality [3,18,23], transcending ethnic boundaries. Of our respondents, 67.9% had not received any vaccine dose to prevent HPV infection and only 37.1% had ever had a cervico-vaginal cytological examination.

Multiple factors have been identified that affect the implementation of a screening program: the availability of healthcare funds in each region or country, medical and economic infrastructure, and social risk perception and tolerance [24]. Despite many efforts to implement screening programs, their success primarily depends on sufficient population coverage. Unfortunately, many countries report suboptimal participation in screening programs [25].

Studies conducted in different countries, that assessed the level of awareness and knowledge of the population about HPV, cervical cancer screening programs, and vaccination programs, noted that low coverage is directly affected by the target population [25,26,27,28,29,30,31,32,33]. These studies also highlighted the importance of healthcare providers, general practitioners, and gynecologists, in both opportunistic screening and organized programs [34,35].

It is estimated that three lifetime tests, between the ages of 30 and 45, reduce the risk of CC by up to 50%. Attention to symptoms and rapid access to healthcare services allow for increased survival with CC [36]. HPV infection is an important condition but is insufficient for the development of CC. Numerous cofactors have been identified: first sexual intercourse at an early age, infections caused by *Herpes simplex* virus Type 2 or *Chlamydia trachomatis*, frequent change of sexual partners, long-term use of oral contraceptives, smoking, low intake of fruits and vegetables, obesity, multiple pregnancies, childbirth before the age of 17, and a family history of CC [27,37,38].

Of the respondents included in our study, 572 had started their sex life under the age of 18; of the women who had ever been tested for HPV infection, about 2/3 had started their sex life under 18. Most had only one sexual partner, and regarding the contraceptive methods used, 35.7% of Group 1 women were unaware of or used any contraceptive method, while 32.2% used hormonal contraceptives. Other studies have also shown that the vast majority of Roma women have one or two sexual partners in their lifetime [14,39].

Of the women who have never been tested for HPV infection, 63.8% do not know/use any contraceptive method (Table 2). Most of the respondents do not consume alcohol or smoke; however, among those who have been tested for HPV, 18.7% smoke an average of 11–20 cigarettes a day, and 18.3% use alcohol 1–2 times a week. Mixed nutrition predominates in women who have been tested. Regarding the history of CC, 8.7% of women who had ever been tested for HPV had a diagnosis of CC. Within the families of these women, 17.3% have a family member diagnosed with CC, while 28.4% have a family member with another type of cancer. Regarding pregnancy, 36.3% of the tested women have had over 5 pregnancies, compared to 30.6% of the untested ones who have had over 2–3 pregnancies. As such, of the known risk factors, only a part was identified in the Roma population surveyed in our study: first sexual intercourse at an early age, multiple pregnancies, childbirth before the age of 17, and a family history of CC.

In Cambrea et al.’s study [17], the correlation of HPV with sexual activity (number of sexual partners and age at first intercourse) was followed in a group of 40 women. The risk of having HPV was 4.13 times higher in the group of women with more than two sexual partners compared to the group of women with one or two sexual partners (*p* < 0.05). The risk of a woman who started her sex life before the age of 18 years to develop an HPV infection was 4.55 times higher compared to a woman who started her sex life after the age of 18 years (*p* < 0.05).

Abstaining from sexual activity (for example, abstaining from any genital contact with another person) is the safest way to prevent HPV infection. Individuals can also reduce their chances of becoming infected with HPV by being in a monogamous relationship with a partner, limiting their number of sexual partners, and choosing a partner who has had no or few previous sexual partners. However, even people with only one sexual partner can be infected with HPV. Consistent and correct use of a condom can reduce the risk of HPV and HPV-associated diseases (e.g., genital warts and CC).

According to data from our study, of the women ever tested for HPV infection, 9.7% were detected with this virus. The study of Ilisiu et al. [40] on screening for CC hr-HPV analyzed the results of 2060 women aged 18 to 70 years. The highest prevalence rates of hr-HPV were observed in Romanian women (17.9%; 95 CI: 15.5–20.7%), Hungarian ethnicity (16.6%; 95% CI: 13.1–20.8%), Russian women (15.6%; 95% CI: 11.3–21.3), and women living in urban areas (20.0%; 95 CI: 18.5–28.0%). Hr-HPV prevalence rates were lower for the Roma population (7.8%; 95% CI: 4.7–12.5%). This study also mentions that a higher proportion of Romanian women were tested in public hospitals (43.4%), while 71.0% of Roma women were tested by their general practitioners.

In our study, the majority of women were registered with a GP (97.1%), and 25.3% had had surgery for gynecological conditions.

A serious problem identified in our study was the lack of information about anything that means HPV infection and CC: symptoms, prevention through screening programs, vaccination, etc. The survey also asked the question: *How much do you know about CC and how can it be prevented*. Most answers were obtained from the group of women tested, but the percentage of positive answers did not exceed 7%: *CC can also develop because of HPV, we must have at least 1 Pap test per year, but I do not know exactly how we can prevent it* (2.4%); *about CC I know that when you bleed between menses or during sexual intercourse or you have great pain during sexual intercourse, or if you have an abundant discharge with blood veins, then you must see a gynecologist* (2.4%); *it is deadly, we must have regular check-ups; it is transmitted genetically* (3.8%); *transmission through sexual intercourse* (2.4%); *we must see a specialist* (3.8%); *if it is in an advanced stage, it cannot be treated* (3.8%); and *it can be prevented by the vaccine* (6.2%). Of the women tested, 63% do not know much about CC, as opposed to 85.7% of the group of untested women.

A cross-sectional study, conducted through a structured questionnaire among both Roma and Romanian women, revealed that the primary obstacle hindering their participation in screening was a lack of awareness regarding the existence of the screening program. Additionally, financial constraints, especially the inability to cover expenses associated with positive test results, and exclusion from the free screening program in Romania were significant barriers faced by both ethnic groups [39].

In Simion et al.’s study [16], a group of 528 women responded to a questionnaire assessing demographic data, their level of knowledge about HPV screening and vaccination, and their access to cervical-cancer-specific medical services. Respondents were asked if they had previously had a Pap test: 21% of women had never had a Pap test in their life. Of the 384 women with a previous Pap smear, 77 were tested in the previous year, 137 women had a Pap smear 2 to 4 years before, and for 170 women, it was 5 years or more since the test. For 73 of these women, it was more than 10 years since the test, while it was more than 20 years for 26 women.

According to this study, the most frequent reasons cited for the lack of intention to perform a Pap test were, in order of frequency: lack of time, lack of financial means, the absence of any symptomatology, lack of information about where/how to be tested, and fear of the procedure/test results. Also, we identified that most of the respondents know that there is an effective vaccine against HPV, but only 4% are vaccinated.

A study on (non)participation of Roma women in CC screening in Romania found major differences in screening perceptions between users and providers. The study recommended that, to improve attendance, all women should be involved in planning, mobilizing, implementing, and evaluating the program to build trust between those offering screening and potential participants [15].

The proportion of women with a high level of education who undergo screening is almost five times higher than that of women with a low level of education. In terms of income, the proportion of high-income women receiving a screening test is three times higher than that of low-income women (43.3% vs. 12.8%) [41].

It is known that Roma women participate less often in screening programs compared to non-Roma women [14,15], and this aspect is related to a lower level of knowledge about the existence of screening programs, distrust in the policy of free screening programs, and the benefits that screening could bring [38].

Previous studies based on questionnaires addressed to both Romanian populations and ethnic minorities in Romania have determined a low participation in screening, which is mainly caused by a lack of awareness of the existence of the screening program and a perceived lack of financial means to be examined and follow positive test results [20,40].

The Roma community is subject to its own rules of life, which are often not known/shared by the majority population, which can lead to misunderstandings and even conflicts between communities. The use of health mediators, who are aware of these rules and who have the knowledge and means to address the Roma population, can facilitate the access of Roma women to screening programs. An important role would also be played by the GP because our study also found that when Roma women are tested for CC, they do so with a GP rather than at a public or private hospital.

Limitations of our study: One limitation of our study is the recruiting approach that was used, primarily through an online survey in which respondents self-declared their Roma ethnicity, with these being the only eligible subjects to be included, which could introduce bias, since it may not properly represent the Roma community’s variety, including different traits and living conditions. However, the use of a relatively large sample should have partially addressed this potential selection bias. Secondly, some respondents who completed the questionnaire may have been reserved in giving correct answers, especially to questions about sexual activity, although they were assured of the anonymity and confidentiality of their answers. Moreover, the questionnaires that did not have complete responses were removed from the statistical analysis.

The practical implications of this study’s findings for public health initiatives and policy reforms are important. It emphasizes the necessity of establishing a safe and nonjudgmental setting for data collection, particularly when addressing sensitive subjects like sexual activity.

## 5. Conclusions

CC is a priority area of public health in Romania, as it is one of the national chronic conditions that can be diagnosed/treated at an early stage, having a greater impact on poorer and vulnerable people, such as the Roma female population. Access to primary health and prevention services (screening) aimed at prevention, early detection, diagnosis, and treatment of precancerous lesions of the cervix, including the most vulnerable segments of populations living in more precarious socioeconomic conditions and with less accessibility to healthcare services, contributes to the promotion of social cohesion through health. Given the multitude of social disadvantages faced by the Roma community, there is an urgent need for comprehensive information campaigns concerning the risk of CC. These campaigns should be conducted at the individual, group, and community levels, aiming to ensure that women have a thorough understanding of the disease’s underlying causes and are aware of the available preventive measures. Such information campaigns should adopt a sustainable, equitable, and optimized approach, with the primary goal of educating and raising awareness among vulnerable populations regarding their specific healthcare needs, patient rights, and the vital importance of life-saving screening tests.

## Figures and Tables

**Figure 1 medicina-59-01885-f001:**
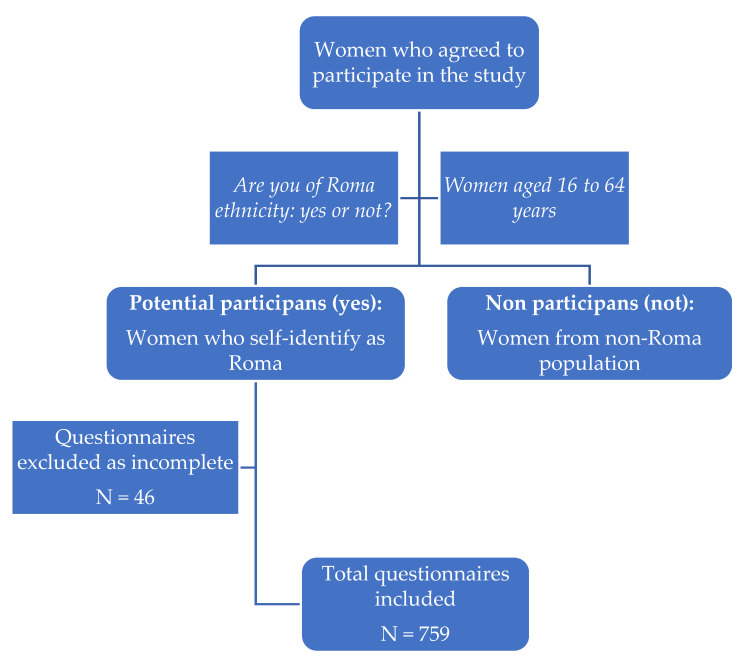
Flowchart showing steps for inclusion/exclusion.

**Table 1 medicina-59-01885-t001:** Demographic characteristics of the study group by HPV testing.

Variables		Have You Ever Been Tested for HPV?		*p*-Value
		Group 1 Yes (289)	Group 2 No (470)	
What age group are you in?	<18 years	7 (2.4)	129 (27.4)	0.0001
	19–24 years	53 (18.3)	73 (15.5)	
	25–34 years	75 (26.0)	171 (36.4)	
	35–44 years	73 (25.3)	43 (9.1)	
	45–54 years	60 (20.8)	39 (8.3)	
	55–64 years	21 (7.3)	15 (3.2)	
Origin	Rural	154 (53.3)	359 (76.4)	0.0001
Education	No schooling	29 (10)	84 (17.9)	0.0001
	Secondary education	98 (33.9)	306 (65.1)	
	High school education	148 (51.2)	80 (17.0)	
	Undergraduate education	14 (4.8)	0 (0.0)	
Marital status	Married	202 (69.9)	183 (38.9)	0.0001
	Unmarried	66 (22.8)	279 (59.4)	
	Divorced	14 (4.8)	0 (0.0)	
	Widow	7 (2.4)	8 (1.7)	
How many cigarettes do you smoke on average per day?	Under 5/day	28 (9.7)	43 (9.1)	0.0001
	5–10/day	21 (7.3)	91 (19.4)	
	11–20/day	54 (18.7)	82 (17.4)	
	Over 20/day	18 (6.2)	8 (1.7)	
	I do not smoke	168 (58.1)	246 (52.3)	
Do you use alcohol?	1–2 times a week	53 (18.3)	0 (0.0)	0.0001
	I do not use alcohol at all	236 (81.7)	470 (100)	
Which of the following eating habits do you employ?	Exclusively meat products	0 (0.0)	36 (7.7)	0.0001
	Exclusively vegetables	0 (0.0)	14 (3)	
	Mixed diet	289 (100)	420 (89.4)	

**Table 2 medicina-59-01885-t002:** Group features according to sex life vs. testing for HPV infection.

Variables		Have You Ever Been Tested for HPV?		*p*-Value
		Group 1 Yes (289)	Group 2 No (470)	
At what age did you start your sexual life?	under 18	184 (63.7)	388 (82.6)	0.0001
	over 18	105 (36.3)	61 (13)	
	I have not started my sex life	0 (0.0)	21 (4.5)	
What is the number of your sexual partners?	1	150 (51.9)	236 (50.2)	0.001
	2–5	118 (40.8)	184 (39.1)	
	Over 5	21 (7.3)	29 (6.2)	
	None	0 (0.0)	21 (4.5)	
What is the frequency of vaginal sex?	Under 3/week	261 (90.3)	373 (79.4)	0.001
	Over 3/week	28 (9.7)	76 (16.1)	
	I have not started my sex life	0 (0.0)	21 (4.5)	
How many pregnancies have you had?	1	37 (12.8)	83 (17.7)	0.0001
	2–3	87 (30.1)	144 (30.6)	
	4–5	25 (8.7)	141 (30)	
	Over 5	105 (36.3)	74 (15.7)	
	None	35 (12.1)	28 (6)	
Of the total number of pregnancies, how many miscarriages have you had?	1	40 (15.7)	80 (18.1)	0.0001
	2–3	50 (19.7)	48 (10.9)	
	Over 3	55 (21.6)	33 (7.5)	
	None	109 (42.9)	281 (63.5)	
What contraceptive methods do you use?	Contraceptive barriers	11 (3.8)	21 (4.5)	0.0001
	Coitus interruptus	7 (2.4)	0 (0.0)	
	Hormonal contraceptives	93 (32.2)	62 (13.2)	
	Intrauterine devices	33 (11.4)	14 (3)	
	Calendar method	42 (14.5)	7 (1.5)	
	I do not know of/use any contraceptive method	103 (35.7)	300 (63.8)	
	Sterilization	0 (0.0)	66 (14)	

**Table 3 medicina-59-01885-t003:** Knowledge about CC, genital tract infections, diagnosis, treatment, and prevention.

Variables		Have You Ever Been Tested for HPV?		*p*-Value
		Group 1 Yes (289)	Group 2 No (470)	
Are you registered with a general practitioner?	Yes	289 (100)	448 (95.3)	0.01
	No	0 (0.0)	22 (4.7)	
Have you ever had a CC diagnosis?	Yes	25 (8.7)	0 (0.0)	0.0001
	No	264 (91.3)	470 (100)	
Is there anyone in your family with a CC diagnosis?	Yes	50 (17.3)	50 (10.6)	0.008
	No	239 (82.7)	420 (89.4)	
Do you have any family members with other types of cancer?	Yes	82 (28.4)	64 (13.6)	0.0001
	No	207 (71.6)	406 (86.4)	
Have you ever undergone gynecological examinations?	Yes	289 (100)	387 (82.3)	0.0001
	No	0 (0.0)	83 (17.7)	
If so, how long ago?	Within the last year	218 (75.4)	257 (54.7)	0.0001
	Within the last 2 years	21 (7.3)	50 (10.6)	
	Within the last 5 years	50 (17.3)	80 (17)	
	Never	0 (0.0)	83 (17.7)	
Have you ever heard of HPV?	Yes	182 (63.0)	140 (28.8)	0.0001
	No	107 (37.0)	330 (70.2)	
Have you been vaccinated against HPV?	Yes, with at least one dose	47 (16.2)	91 (19.3)	0.002
	Yes, with 2–3 doses	57 (19.7)	48 (10.2)	
	No	185 (64.1)	331 (70.5)	
Have you ever had a cervico-vaginal cytological examination?	Yes	282 (97.6)	0 (0.0)	0.0001
	No	7 (2.4)	470 (100)	
Following the Pap test, what types of cervical intraepithelial lesions were found?	Cancerous cells	11 (3.8)	0 (0.0)	0.0001
	HSIL type cells	7 (2.4)	0 (0.0)	
	LSIL type cells	7 (2.4)	0 (0.0)	
	Inflammatory cells	21 (7.3)	0 (0.0)	
	Normal cells	201 (69.6)	0 (0.0)	
	I do not know	35 (12.1)	0 (0.0)	
	No such testing	7 (2.4)	470 (100)	
If you have undergone treatment, was the lesion treated?	Yes	56 (19.4)	0 (0.0)	0.0001
	No	18 (6.2)	0 (0.0)	
	No treatment	215 (74.4)	470 (100)	
Have you ever had a genital tract infection?	Yes	181 (62.6)	264 (56.2)	0.079
	No	108 (37.4)	206 (43.8)	
If so, what infection did you have?	Candidiasis	63 (21.8)	43 (9.2)	0.0001
	HPV	28 (9.7)	0 (0.0)	
	Urinary infection	146 (50.5)	250 (53.2)	
	Vaginitis	7 (2.4)	7 (1.5)	
	No infection	108 (37.4)	206 (43.8)	
Have you ever had surgery for gynecological diseases?	Yes	85 (29.4)	107 (22.8)	0.048
	No	204 (70.6)	363 (77.2)	

## Data Availability

Not applicable.

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
