# Peer review of "Assessing the Level of Knowledge and Experience Regarding Cervical Cancer Prevention and Screening among Roma Women in Romania"

_medicina, 2023, doi:10.3390/medicina59101885_

Round 1

Reviewer 1 Report

Authors conducted the questionnaire based study “Assessing the Level of Knowledge and Experience Regarding Cervical Cancer Prevention and Screening among Roma Women in Romania”. A cross-sectional study was completed using a self-administered questionnaire applied only to Roma women in Romania to identify correct information of the about the risk of Cervical cancer as well as the importance of understanding the causes of the disease and awareness of the available prevention methods I have enjoyed learning about the presented results as the study is interesting, but it has many limitations, more robust appropriate statistical analysis is needed, and the methods section is incomplete. Finally, the results are not representative of the whole population.

Comment 1: Abstract:

Authors should improve the abstract in following aspects.

Several sentences are fragmented and lack clarity. For example, "Romania ranks among the countries with a notably high rate of preventable deaths due to inadequacies in prevention, screening, early detection, and timely management processes" could be rephrased for better readability.

While the abstract mentioned statistics (e.g., percentages), it would be more informative to include specific numerical values i.e. confidence interval, especially in the Results section.

Comment 2: Introduction:

The introduction is lengthy and lacks clear subsections, making it challenging to follow. Authors are advised to consider organizing it into sections addressing different aspects of the background, healthcare challenges, and cervical cancer. There are punctuation and grammatical errors throughout the section.

Comment 3: Materials and Methods:

 This section is incomplete. Include study design. Include the STROBE check list (as an Appendix), and verify if they are complying with all the items on the STROBE check list. What were the inclusion and exclusion criteria? Explain if the questionnaire was pilot tested. Have the participants completed an informed consent? It remains to include the study population, the sample and the sampling.

Comment 4: Results:

The results mentioned of statistical significance is made without specific p-values or confidence intervals and adjusted ODD ratios must be calculated and incorporated where applicable. Providing these values would enhance the validity of the results.

Comment 5: Discussion:

The Discussion section should offer a thorough interpretation of the results and relate them to the research objectives.

Comparing the findings of this study with previous research or national statistics would add context to the discussion. Authors must include a paragraph describing the public health implications.

 Comment 6: The limitations of the study should be discussed, including potential sources of bias or error.

What are the practical implications of the study's findings, and how might they inform public health interventions or policy changes?

Comment 7: Conclusions:

The conclusion briefly summarizes the main findings but could be more comprehensive.

Minor English Language editing is required in term of grammar and punctuations used throughout the manuscript

Author Response

Comments and Suggestions for Authors

Authors conducted the questionnaire based study “Assessing the Level of Knowledge and Experience Regarding Cervical Cancer Prevention and Screening among Roma Women in Romania”. A cross-sectional study was completed using a self-administered questionnaire applied only to Roma women in Romania to identify correct information of the about the risk of Cervical cancer as well as the importance of understanding the causes of the disease and awareness of the available prevention methods I have enjoyed learning about the presented results as the study is interesting, but it has many limitations, more robust appropriate statistical analysis is needed, and the methods section is incomplete. Finally, the results are not representative of the whole population.

Response to Reviewer 1 Comments

We thank the reviewer for the comments.

Comment 1: Abstract:

Authors should improve the abstract in following aspects.

Several sentences are fragmented and lack clarity. For example, "Romania ranks among the countries with a notably high rate of preventable deaths due to inadequacies in prevention, screening, early detection, and timely management processes" could be rephrased for better readability.

We rephrased.

While the abstract mentioned statistics (e.g., percentages), it would be more informative to include specific numerical values i.e. confidence interval, especially in the Results section.

In this study for the statistical analysis we used a bivariate analysis, i.e. the chi square test which compared the percentage differences between various variables of the two groups. This test achieved statistical significance in most comparisons. It was not necessary to calculate various risks, such as OR (it was not even possible in many situations) and implicitly neither the confidence interval.

Comment 2: Introduction:

The introduction is lengthy and lacks clear subsections, making it challenging to follow. Authors are advised to consider organizing it into sections addressing different aspects of the background, healthcare challenges, and cervical cancer. There are punctuation and grammatical errors throughout the section.

We revised the introduction section to enhance readability and coherence, while also addressing all grammatical errors.

Comment 3: Materials and Methods:

 This section is incomplete. Include study design. Include the STROBE check list (as an Appendix), and verify if they are complying with all the items on the STROBE check list. What were the inclusion and exclusion criteria? Explain if the questionnaire was pilot tested. Have the participants completed an informed consent? It remains to include the study population, the sample and the sampling.

We gave details about the way of sampling and surveying the target population, also respecting the Strobe criteria.

Comment 4: Results:

The results mentioned of statistical significance is made without specific p-values or confidence intervals and adjusted ODD ratios must be calculated and incorporated where applicable. Providing these values would enhance the validity of the results.

We relied on the bivariate analysis, not considering it necessary to calculate the OR, respectively the confidence interval. We applied the chi square test which compares the difference between the percentages of the variables in the two groups.

Comment 5: Discussion:

The Discussion section should offer a thorough interpretation of the results and relate them to the research objectives.

Comparing the findings of this study with previous research or national statistics would add context to the discussion. Authors must include a paragraph describing the public health implications.

New information has been added in the Discussion section regarding the national statistics.

 Comment 6: The limitations of the study should be discussed, including potential sources of bias or error.

What are the practical implications of the study's findings, and how might they inform public health interventions or policy changes?

Limitation about bias and also the practical implications are now added in the Discussion section.

Comment 7: Conclusions:

The conclusion briefly summarizes the main findings but could be more comprehensive.

Comments on the Quality of English Language

Minor English Language editing is required in term of grammar and punctuations used throughout the manuscript.

We thank the reviewer for the recommendation. We have thoroughly reviesed the manuscript to address this issue.

Reviewer 2 Report

Review report for authors:

 A brief summary:

This was a cross-sectional study , and it has been conducted using a self-administered questionnaire applied only to Roma women in Romania. The authors enrolled 759 patients in the study. These were divided into 2 groups, Group 1 comprised  (38.1%) women who had been tested for HPV infection, while Group 2 included  (61.9%)  women who had never been tested for HPV infection.

As the authors stated in the text of the introduction itself the purpose of this study was to ascertain the accuracy and consistency of information within the Roma population concerning the risk of cervical cancer (CC). The authors underscore the significance of comprehending the underlying causes of the disease and fostering awareness regarding the available prevention methods.

The authors concluded that cervical cancer (CC) continues to be a public health issue in Romania.  Promotion campaigns for HPV vaccination and CC screening are necessary to reduce the associated mortality and morbidity in Romania.

General concept comments:

As the authors of this study already said, study has some limitations:  this was a cross sectional  study conducted using a self-administrated questionnaire applied only to Roma women in Romania.

There is no control group in the search, the study is based on a comparison of a self-fulfilling questionnaire, there is also no comparison with similar studies in Romania as well as in other EU countries. The tested population itself is a limiting factor. In my opinion this is a review professional paper useful only in a narrow medical population that deals with this problem of women in the area.

These are limitations that you as the authors mentioned, but at the same time that are also clear limitations of the research.

The title is clear, well written and concise. It clearly points to the issue that will be discussed in the following text.

The summary is well written and contains all the important results and conculusions of the study.

The introduction  is clearly and interestingly written, it provides an appropriate overview and introduction to the main topic of the paper.

The methods are examplary written but they are not eproducible. The conducted survey questionnaires and diagnostic methods are not so much clearly demonstrated.

The paper is extremely easy to read, with interesting topics, with a clear indication of the flaws of the system that result in devastating public health problems. But the concept of study with filling out questionnaires and sorting on the basis of one variable, without any meaningful control group does not make scientific sense in my opinion.

The results are adequately presented. The results include only tables. The visual disadventage is the absence of graphs. Nevertheless, the results are easy to interpret and understand.

The discussion is clear and concise, i think it is a significant flaw that the authors did not compare their results with similar studies conducted both in Romania and other EU countries. The statements in the discussion are drawn coherently and are supported by appropriate  recent citations.

The literature is correctly cited. Slightly more than 70,8 % of references (17/24) are references of recent publications.

I think that the work should be supported with more recent quotes of similar studys conducted both in Romania and other EU countries.

The conclusion is consistent, concise with clearly presented arguments.

The hypothesis of the article is clearly written, explained but  later  poorly developed methodologically. Experimental design of this study is not truly appropriate for the testing  inicial hypothesis of the study.

In conclusion, I belive that this is an interesting work with a low scientific contribution, and it is  applicable in a narrower, strictly defined gynecological and epideriological  population of medical workers.

I think that the study would be scientifically appropriate if the concept of testing variables were changed and there was a control group, in this case other Romanian women, since  only the Roma women population itself was tested.

Author Response

Comments and Suggestions for Authors

Review report for authors:

 A brief summary:

This was a cross-sectional study , and it has been conducted using a self-administered questionnaire applied only to Roma women in Romania. The authors enrolled 759 patients in the study. These were divided into 2 groups, Group 1 comprised  (38.1%) women who had been tested for HPV infection, while Group 2 included  (61.9%)  women who had never been tested for HPV infection.

As the authors stated in the text of the introduction itself the purpose of this study was to ascertain the accuracy and consistency of information within the Roma population concerning the risk of cervical cancer (CC). The authors underscore the significance of comprehending the underlying causes of the disease and fostering awareness regarding the available prevention methods.

The authors concluded that cervical cancer (CC) continues to be a public health issue in Romania.  Promotion campaigns for HPV vaccination and CC screening are necessary to reduce the associated mortality and morbidity in Romania.

Response to Reviewer 2 Comments

We thank the reviewer for the comments.

General concept comments:

As the authors of this study already said, study has some limitations:  this was a cross sectional  study conducted using a self-administrated questionnaire applied only to Roma women in Romania.

There is no control group in the search, the study is based on a comparison of a self-fulfilling questionnaire, there is also no comparison with similar studies in Romania as well as in other EU countries. The tested population itself is a limiting factor. In my opinion this is a review professional paper useful only in a narrow medical population that deals with this problem of women in the area.

These are limitations that you as the authors mentioned, but at the same time that are also clear limitations of the research.

The title is clear, well written and concise. It clearly points to the issue that will be discussed in the following text.

The summary is well written and contains all the important results and conclusions of the study.

The introduction is clearly and interestingly written, it provides an appropriate overview and introduction to the main topic of the paper.

The methods are examplary written but they are not eproducible. The conducted survey questionnaires and diagnostic methods are not so much clearly demonstrated.

The paper is extremely easy to read, with interesting topics, with a clear indication of the flaws of the system that result in devastating public health problems. But the concept of study with filling out questionnaires and sorting on the basis of one variable, without any meaningful control group does not make scientific sense in my opinion.

The results are adequately presented. The results include only tables. The visual disadventage is the absence of graphs. Nevertheless, the results are easy to interpret and understand.

The discussion is clear and concise, i think it is a significant flaw that the authors did not compare their results with similar studies conducted both in Romania and other EU countries. The statements in the discussion are drawn coherently and are supported by appropriate  recent citations.

The literature is correctly cited. Slightly more than 70,8 % of references (17/24) are references of recent publications.

I think that the work should be supported with more recent quotes of similar studys conducted both in Romania and other EU countries.

The conclusion is consistent, concise with clearly presented arguments.

The hypothesis of the article is clearly written, explained but  later  poorly developed methodologically. Experimental design of this study is not truly appropriate for the testing  inicial hypothesis of the study.

In conclusion, I belive that this is an interesting work with a low scientific contribution, and it is  applicable in a narrower, strictly defined gynecological and epidemiological  population of medical workers.

I think that the study would be scientifically appropriate if the concept of testing variables were changed and there was a control group, in this case other Romanian women, since only the Roma women population itself was tested.

We revised the introduction section to enhance readability and coherence, while also addressing all grammatical errors.

New information has been added in the Discussion section regarding the national statistics. Limitation about bias and also the practical implications are now added in the Discussion section. We thank the reviewer for the recommendation. We have thoroughly reviesed the manuscript to address this issue. The study is part of a research project, it focuses only on Roma women, because studies for non-Roma women were also carried out within the project. By reviewing the article we made comparisons with national and international studies.

Round 2

Reviewer 1 Report

Authors have diligently addressed almost all the comments and concerns raised during the review process. The revisions made have significantly improved the quality and clarity of the article. However, there are still need to improve the methodology section. It would be more appropriate to add inclusion/exclusion criteria based upon multiple factors. I recommend to add graphical presentation of inclusion & Exclusion criteria. Although, authors have added this section only depending upon the nationality. Moreover, authors should recheck the references cited as literature has been updated in the text.

Author Response

Authors have diligently addressed almost all the comments and concerns raised during the review process. The revisions made have significantly improved the quality and clarity of the article. However, there are still need to improve the methodology section. It would be more appropriate to add inclusion/exclusion criteria based upon multiple factors. I recommend to add graphical presentation of inclusion & Exclusion criteria. Although, authors have added this section only depending upon the nationality. Moreover, authors should recheck the references cited as literature has been updated in the text.

We have completed the article with the inclusion and exclusion criteria of the participants in the study, upon recommendation we have introduced a flowchart that shows the steps for inclusion and exclusion. We have checked and corrected the cited references.

Reviewer 2 Report

In my opinion, I think that research has been greatly improved, especially when it comes to study comparisons. The research is interesting for the population of both epidemiologists and gynecologists who are specifically engaged in this issue. This research opens up opportunities for improving this and similar other studies of the said population, in this major public problem of the listed population of women.

The authors may possibly consider listing the limitations of the study and what else could be done as a recommendation in subsequent studies by the author, but the same is not necessarily crucial.

Author Response

In my opinion, I think that research has been greatly improved, especially when it comes to study comparisons. The research is interesting for the population of both epidemiologists and gynecologists who are specifically engaged in this issue. This research opens up opportunities for improving this and similar other studies of the said population, in this major public problem of the listed population of women.

The authors may possibly consider listing the limitations of the study and what else could be done as a recommendation in subsequent studies by the author, but the same is not necessarily crucial.

We have completed the article with the inclusion and exclusion criteria of the participants in the study, upon recommendation we have introduced a flowchart that shows the steps for inclusion and exclusion. We have checked and corrected the cited references.
